# Beyond Conventional Antifungals: Combating Resistance Through Novel Therapeutic Pathways

**DOI:** 10.3390/ph18030364

**Published:** 2025-03-04

**Authors:** Helal F. Hetta, Tameem Melhem, Hashim M. Aljohani, Ayman Salama, Rehab Ahmed, Hassabelrasoul Elfadil, Fawaz E. Alanazi, Yasmin N. Ramadan, Basem Battah, Michelangelo Rottura, Matthew Gavino Donadu

**Affiliations:** 1Division of Microbiology, Immunology and Biotechnology, Department of Natural Products and Alternative Medicine, Faculty of Pharmacy, University of Tabuk, Tabuk 71491, Saudi Arabia; rahmed@ut.edu.sa (R.A.); habdelgadir@ut.edu.sa (H.E.); 2Third Faculty of Medicine, Charles University, Ruská 87, 100 00 Prague, Czech Republic; tameem.melhem@lf3.cuni.cz; 3Department of Clinical Laboratory Sciences, College of Applied Medical Sciences, Taibah University, Madina 41477, Saudi Arabia; hsnani@taibahu.edu.sa; 4Department of Pathology and Laboratory Medicine, College of Medicine, University of Cincinnati, Cincinnati, OH 45221, USA; 5Department of Pharmaceutics, Faculty of Pharmacy, University of Tabuk, Tabuk 71491, Saudi Arabia; agrawan@ut.edu.sa; 6Department of Pharmacology and Toxicology, Faculty of Pharmacy, University of Tabuk, Tabuk 71491, Saudi Arabia; falnazi@ut.edu.sa; 7Department of Microbiology and Immunology, Faculty of Pharmacy, Assiut University, Assiut 71515, Egypt; yasmine_mohamed@pharm.aun.edu.eg; 8Department of Biochemistry and Microbiology, Faculty of Pharmacy, Antioch Syrian Private University, Maaret Saidnaya 22734, Syria; basem.battah.sc@hotmail.com; 9Department of Clinical and Experimental Medicine, University of Messina, 98125 Messina, Italy; mrottura@unime.it; 10Hospital Pharmacy, Giovanni Paolo II Hospital, ASL Gallura, 07026 Olbia, Italy; 11Department of Medicine, Surgery and Pharmacy, Scuola di Specializzazione in Farmacia Ospedaliera, University of Sassari, 07100 Sassari, Italy

**Keywords:** antifungal resistance, MDR, future strategies, nanotechnology

## Abstract

The rising burden of fungal infections presents a significant challenge to global healthcare, particularly with increasing antifungal resistance limiting treatment efficacy. Early detection and timely intervention remain critical, yet fungal pathogens employ diverse mechanisms to evade host immunity and develop resistance, undermining existing therapeutic options. Limited antifungal options and rising resistance necessitate novel treatment strategies. This review provides a comprehensive overview of conventional antifungal agents, their mechanisms of action, and emerging resistance pathways. Furthermore, it highlights recently approved and investigational antifungal compounds while evaluating innovative approaches such as nanotechnology, drug repurposing, and immunotherapy. Addressing antifungal resistance requires a multifaceted strategy that integrates novel therapeutics, enhanced diagnostic tools, and future research efforts to develop sustainable and effective treatment solutions.

## 1. Introduction

Fungi are ubiquitous eukaryotic organisms present in soil, plants, air, surfaces, and even within the human body. They exist as unicellular (yeasts) or multicellular (molds) organisms, with some species exhibiting dimorphic characteristics. Functionally, fungi play a crucial role in ecological nutrient cycling and have substantial biotechnological applications, including antibiotic production, fermentation, and bioengineering [1,2,3,4,5]. However, while many fungi are beneficial, others pose significant risks due to their ability to cause disease in plants, animals, and humans. Pathogenic fungi employ diverse mechanisms to evade host immune responses, resulting in persistent and potentially life-threatening infections [6].

Fungal infections are one of the major public health issues since they are linked to life-threatening mycoses and death [7,8,9]. More than 300 million individuals worldwide are affected by severe fungal infections each year [10]. In 2017, fungal infections were responsible for more than 75,000 hospital admissions and about $7.2 billion in medical expenses in the United States [11]. The CDC categorizes fungal illnesses into four groups: common fungal infections, region-specific infections, infections affecting immunocompromised individuals, and other health conditions related to fungi (Figure 1) [7]. Based on another categorization, according to the affected site of infection, fungal infections may include superficial, cutaneous, subcutaneous, mucosal, as well as systemic infections [12].

Superficial, cutaneous, subcutaneous, and mucosal fungal infections affect the skin, mucous membranes, and keratinous tissues, leading to conditions such as oral, pharyngeal, or vaginal thrush; athlete’s foot; and ringworm (tinea) [13].

Systemic infections, also known as invasive infections, are caused by fungi that penetrate deep tissues and eventually pose a life-threatening risk and cause diseases like invasive candidiasis and aspergillosis [14].

Despite effective antifungal medications, fungal infections often have a higher mortality rate than malaria and tuberculosis [15]. So, new technologies and strategies are urgently needed to control this serious issue.

Despite advancements in antifungal therapy, critical research gaps persist. First, the molecular mechanisms driving multidrug resistance in emerging fungal pathogens such as *Candida auris* and *Aspergillus fumigatus* require further investigation. Second, the development of novel antifungal agents with unique mechanisms of action remains a priority, particularly given the increasing resistance to conventional azoles and echinocandins.

This review expands on prior studies by exploring emerging therapeutic strategies such as nanotechnology and antifungal lock therapy. We also discuss emerging fungal pathogens and their evolving resistance patterns, which have not been extensively covered in earlier studies.

## 2. Driving Forces for the Development of Fungal Infection and Resistance

Some fungi live on the skin and in the digestive tract as commensal organisms, and they are considered risk factors for several fungal infections if they are delocalized from their natural habitat. For example, the rising incidence of fungal infections is linked to factors such as excessive antibiotic use, chemotherapy, and immunosuppressive treatments, which disrupt microbial balance and weaken host defenses [16,17]. Moreover, invasive medical devices, including catheters and transplantation procedures, create entry points for fungal pathogens, increasing the risk of invasive disease [18,19]. Recent health crises, including seasonal influenza outbreaks and the COVID-19 pandemic, have exacerbated the severity and frequency of fungal infections [20,21,22].

Therapeutic failure and resistance arise when patients do not respond to antifungal medications despite appropriate dosing. Several factors account for this therapeutic failure. For instance, immunocompromised patients are more susceptible to treatment failure because the antifungal medication is not supported by a strong immunological response in the battle against an infection [23]. Also, implanted devices such as indwelling catheters and artificial valves facilitate recurrent infections due to biofilm formation, which hinders drug penetration and eradication [24]. In the case of intra-abdominal infections and abscesses, treatment failure occurs because of the exposure of fungi to a suboptimal concentration of a drug [25]. Also, the emergence of resistance is linked to prolonged, repetitive, or prophylactic drug exposure. The same is true for the use of agricultural fungicides, which have the same molecular receptors as systemic antifungals and have consequently seeded the environmental reservoirs with resistant strains [26,27]. Factors such as delayed diagnosis, poor compliance, co-infections (e.g., cavitary lesions and abscesses), drug–drug interactions, and obesity contribute to treatment failure.

## 3. Conventional Antifungal Drugs and Their Mechanism of Action

There are approximately five groups of conventional antifungal medications, including azoles, polyenes, echinocandins, allylamines, and pyrimidine analogs, that can be used for topical and systemic antifungal treatments (Figure 2, Table 1) [28,29,30]. More than 200 polyenes with antifungal activity have been found since the 1950s, yet amphotericin B (AmB) is still the only polyene medication of choice for the treatment of invasive fungal infections [31]. Unfortunately, its effectiveness is restricted by frequent and occasionally serious adverse events, particularly nephrotoxicity and infusion reactions [32]. Early in the 1980s, ketoconazole, the first systemic azole, was marketed. A decade later, the first-generation triazoles, fluconazole and itraconazole, were also available. Later on, the second-generation triazole, voriconazole, took the place of AmB as the preferred medication for the treatment of invasive aspergillosis and the majority of other filamentous fungal infections due to its superior efficacy and reduced toxicity [33]. In the 2000s, echinocandins were introduced as first-line medicines for the treatment of invasive candidiasis [34]. The antimetabolite 5-flucytosine (5-FC) was initially developed and utilized in the treatment of cancer, and it is also taken in combination with AmB, mainly for the treatment of cryptococcal meningitis [35,36].

### 3.1. Polyenes

They include AmB and nystatin. They bind to and interact with the surface of the fungal cell membrane, ergosterol, through hydrophobic interaction as well as lead to the formation of hydrophilic pores and increase the permeability of ions via the subsequent change in ion gradient inside and outside the cell membrane and via the loss of cell integrity, finally leading to fungal cell death [37]. The interaction of AmB with ergosterol in the fungal cell membrane creates subunit oligomers of membrane-permeabilizing ion channels that encourage intracellular K^+^ and Mg^2+^ leakage with a reciprocal influx of Na^+^ and Ca^2+^ ions, which have a fatal impact on fungal cells [38,39]. AmB also acts by another mechanism, through the building up of reactive oxygen species (ROS), and subsequently causes damage to mitochondria, proteins, DNA, and membranes [40,41]. ROS damage the functionality of cell membranes by oxidizing lipid membranes and lipoprotein receptors [42]. Additionally, fungicidal activity may be produced by AmB and ergosterol interactions alone, rather than through pore formation [43,44]. AmB may either cluster around ergosterol at the cell membrane surface to behave as a sponge that removes ergosterol from the cell membrane or it can adhere to and sequester cell membrane ergosterol, resulting in the instability of the cell membrane [45].

Unfortunately, it was proven that when the glomerular filtration rate of polyenes is lowered, it can cause kidney toxicity in people due to the comparable structure of ergosterol and cholesterol in human cell membranes [46,47,48].

Over the past 40 years, lipid formulations of AmB have been developed primarily to reduce nephrotoxicity. Liposomal AmB, also known as LAmB or AmBisome, has emerged to become the most extensively utilized medication among the approved lipid formulations of AmB for the management of invasive fungal infections [49,50,51]. The mode of action of LAmB is dependent on the existence of AmB in the liposome bilayer, chemical constituents of liposome, its binding affinity for fungal cell walls [52], and its capability to target and pass through the cell wall as well as interact with ergosterol in the fungal cell membrane [53]. LAmB shows decreased nephrotoxicity. In the case of LAmB, AmB binds with cholesterol present in the liposomal bilayer, preventing AmB from being released from the liposome and interacting with cholesterol in the mammalian cell membrane, causing nephrotoxicity [54]. Liposomes have a size of 60–80 nm, and the cell wall’s porosity is only approximately 5.8 nm; these findings suggest that there is fast cell wall remodeling that permits liposomes to travel intact through the cell wall to ergosterol in the cell membrane, where it subsequently releases AmB [53,55].

### 3.2. Azoles

The azole class includes triazole and imidazole [56]. Azole drugs bind to and inhibit the heme group of fungal cytochrome p450, which is an enzyme responsible for catalyzing the demethylation step of lanosterol at position 14 and converting it to ergosterol. This demethylation step is important in ergosterol synthesis, which is essential for preserving the integrity and stability of fungal cell membranes. So, the normal function of the fungal cytoplasmic membrane is disrupted by the loss of ergosterol and the formation of poisonous methyl sterols, which have a fungicidal effect on molds and a fungistatic effect on yeasts [57,58,59,60]. The genes *ERG11* in yeasts and *Cyp51* in molds both encode lanosterol 14-alpha demethylase [57].

### 3.3. Echinocandins

The most common clinical echinocandins are caspofungin and micafungin [61]. Echinocandin drugs act by binding to and inhibiting the fungal 1,3 β-D glucan synthase enzyme that is responsible for the synthesis of 1,3 β-D glucan, which is an important component of fungal cell wall construction, leading to the destruction of the fungal cell wall [62,63]. The enzyme 1,3 β-D glucan synthase is encoded by genes from the *FKS* family [63]. Echinocandin drugs possess fungistatic action on *Aspergillus* spp. and fungicidal action on *Candida* spp. but have no action on *Cryptococcus* spp. [64].

Anidulafungin is the third echinocandin antifungal agent to be approved by the US Food and Drug Administration (FDA). Anidulafungin is a semisynthetic lipopeptide derived from *A. nidulana* fermentation products [65]. It disrupts the fungal cell wall by binding to other 1, 3 β-D glucan synthase-like echinocandins [65]. Anidulafungin is not metabolized. Instead, it is slowly degraded by human peptidases and proteases. Additionally, it has a low drug–drug interaction pattern since it does not interact with the cytochrome P450 system [66]. As a result, anidulafungin dosage adjustments depending on gender, age, body weight, illness condition, concurrent medication, or renal or hepatic impairment are not required. Furthermore, it can be combined with other antifungal drugs such as voriconazole and AmB [67,68]. It was reported that echinocandins enhance cardiac toxicity [69,70], but anidulafungin is the least toxic [71]. Anidulafungin liposomes were developed to increase solubility, boost antifungal effectiveness, lessen toxicity, and perhaps minimize susceptibility to resistance. These liposomes show a tendency to connect with *C. albicans* rather than sequestering ergosterol [72].

### 3.4. Allylamines

The most common clinical allylamine drugs are terbinafine and naftifine. They have a broad spectrum of activity and low toxicity. They act by binding to and inhibiting squalene epoxidase, which converts squalene to lanosterol, inhibiting ergosterol synthesis and affecting fungal growth [73]. They are fungicidal against dermatophytes and fungistatic against *Candida* spp. [74].

### 3.5. Pyrimidine Analogs

The most common clinical pyrimidine analog drug is 5-FC. It enters the fungal cells through cytosine permeases where it is deaminated to 5-fluorouracil, which inhibits the synthesis of both nucleic acids (DNA and RNA) and subsequently inhibits protein synthesis. Additionally, 5-FC can cross the blood–brain barrier and can be utilized to treat central nervous system (CNS) fungal infections [75,76]. Unfortunately, 5-FC possesses serious side effects, as it causes bone marrow suppression and nephrotoxicity [77].

### 3.6. Other Antifungal Drugs

Sordarin particularly blocks the protein synthesis elongation cycle [78,79]; griseofulvin inhibits the formation of microtubules [80], and triphenylethylenes suppress calcineurin signaling [81].

## 4. Development of Resistance to Conventional Antifungal Drugs

Generally, the development of resistance to antifungals is categorized into two types: innate (primary) and acquired (secondary). Innate resistance is genetically encoded, naturally occurs in some fungi, and is linked to fungi strains that are naturally resistant to specific antifungals. Acquired resistance occurs because of exposure to a stressful event, most commonly an antifungal medication or its structural mimic [82]. Resistance to antifungals is a major cause of concern because there are so few treatment targets. Three types of systemic antifungals—azoles, echinocandins, and polyenes—have typically been used to effectively treat invasive fungal infections [83]. So, resistance to only one class of medications can seriously reduce the options of available treatments, whereas multidrug resistance may make it very hard to treat fungal infections. It is easier for resistance to emerge when fungi are exposed to sublethal concentrations, as they may dynamically adjust and adapt to changing environmental conditions [84]. The selective pressure of therapeutic, industrial, and agricultural antifungal agent use is reflected in the expansion of drug resistance in pathogenic fungi. More recently, multidrug resistance (MDR) *C. auris* has become a significant global issue for public health due to its potential to result in hospital epidemics with high fatality rates [30,85,86].

### 4.1. Resistance to Azoles

Azoles are the most often utilized antifungal class [87]. Fluconazole is still widely used in the treatment regimen of cryptococcosis and invasive candidiasis, although it is not currently utilized as a first-line treatment [88,89]. Voriconazole, posaconazole, and isavuconazole are commonly used in the treatment of invasive aspergillosis and some other invasive mold infections [90,91].

Azole resistance occurs through three main mechanisms: the overexpression of lanosterol 14-demethylase, alterations in the azole binding site, and the upregulation of multidrug efflux transporters. Different fungal species have different dominant resistance mechanisms, and a single isolate may have more than one active mechanism (Figure 3).

In *Candida* spp., numerous pathways can lead to azole resistance, but the most prevalent pathways include the development of drug exporter pumps and overexpression or point mutations in the *ERG11* gene that encodes the 14-α demethylase enzyme. The overexpression of *ERG11* means an increase in the 14-α demethylase enzyme, azole targets, and ergosterol synthesis. A point mutation in the *ERG11* gene leads to the formation of mutated azole targets with reduced binding affinity [92,93]. The overexpression of the drug exporter pump is the most immediate compensatory reaction that appears as a result of exposure to azoles. Furthermore, two types of exporters are common in *Candida* spp.: the major facilitator superfamily (MFS) exporter pump, which is encoded by the genes *MDR1* and *FLU1*, and the ATP-binding cassette (ABC) exporter pump, which is encoded by the genes *CDR1* and *CDR2*, respectively [94,95]. According to the SENTRY trial, fluconazole resistance is uncommon (3.9% overall), although it is on the rise over time, with more resistance in Latin America than in other regions of the world. Also, it was discovered that every *C. auris* isolate used in this investigation was resistant to fluconazole [96]. It was reported that the *C. glabrata* transcriptional regulator (CgRpn4) is a determinant of azole resistance. RNA sequencing of CgRpn4 upon fluconazole exposure to *C. glabrata* showed that it controls the expression of 212 genes, activating 80 genes and suppressing, most likely indirectly, 132 genes. CgRpn4 directly controls *ERG11* expression, which helps to maintain cell membrane homeostasis and, as a result, reduces azole drug penetration and accumulation [97]. Hexose transporters in *C. glabrata* could be a class of azole importers that have a role in clinical drug resistance in fungal infections and also offer potential targets for effective antifungal treatment. In a recent study, Galocha et al. discovered that the mutation in the CgHxt4/6/7 hexose transporter in *C. glabrata* plays an essential role in azole accumulation and the development of azole resistance [98].

Biofilm formation is another significant drug-resistance mechanism that enhances microbial resistance or promotes other mechanisms like the drug exporter pump [99,100]. The biofilm efficiently lowers the medication level by encasing it in a polymeric matrix rich in glucans [101]. In general, azoles and other systemic antifungals are affected by biofilm construction, which is a universal process.

In *Aspergillus* spp., the most frequent spp. associated with invasive infections, especially in immunodeficient patients, are *A. fumigatus*, *A. flavus*, *A. niger*, and *A. terreus*; they also cause invasive infections [102]. It was reported that patients using triazole prophylaxis have been linked to breakthrough infections caused by non-*fumigatus* spp. of *Aspergillus*, many of which are involved in azole resistance [21,91,103]. These *Aspergillus* spp. are inherently susceptible to triazoles. Two *CYP51* analogs, *CYP51A* and *CYP51B*, are found in *A. fumigatus*, and they share a 63% similarity in their sequences [104]. *A fumigatus’*s acquired resistance to azoles is almost usually linked to alterations in the *CYP51A* gene’s coding sequence, promoter, or both and very infrequently to mutations in *CYP51B* [105]. Also, they may develop resistance through the overexpression of the *CYP51A* gene, so the overproduction of the 14-α demethylase enzyme exceeds the therapeutic concentrations of the azoles [106]. Moreover, the overexpression of the ABC exporter pump is another mechanism of resistance and leads to a decrease in intracellular therapeutic concentration [107]. *Aspergillus* taxonomic research has discovered novel cryptic species that are virtually indistinguishable from one another morphologically by conventional identification techniques. Common *Aspergillus* spp., such as *A. fumigatus*, may complex with cryptic species and form new strains with a higher minimum inhibitory concentration (MIC) to azoles and some AmB types. *A. lentulus*, *A. fumigatiaffinis*, *A. viridinutans*, and *A. pseudofischeri* are examples of complex species of *A. fumigatus* [108]. *A. fumigatus*, as well as *C. albicans*, imports extracellular cholesterol in aerobic circumstances, which enables them to escape the antifungal effects of sterol biosynthesis inhibitors such as azoles [109].

In summary, azole resistance in fungal strains arises from mechanisms such as ERG11 mutations, efflux pump overexpression, and biofilm formation. These factors reduce drug efficacy, particularly against *Candida* and *Aspergillus* spp., complicating treatment. These factors collectively diminish the efficacy of azoles, particularly against *Candida* and *Aspergillus* spp., leading to challenges in antifungal treatment.

### 4.2. Resistance to Echinocandins

Echinocandins are a significant class of antifungal medications, especially as they are used as first-line therapy in invasive candidiasis and as rescue therapy for invasive aspergillosis [110,111]. Echinocandins often still have efficacy against the majority of azole-resistant *Candida* spp. since their mechanism is different from that of azole antifungals. However, over time, resistance to echinocandins is increasing dramatically, particularly in isolates of *C. glabrata*. For instance, 8% to 9% of *C. glabrata* bloodstream isolates gathered between 2009 and 2010 for the SENTRY antimicrobial surveillance program were found to be caspofungin resistant [112,113,114].

In *Candida* spp., especially in *C. glabrata*, echinocandin resistance is most frequently caused by mutations, through amino acid substitutions, in the *FKS-1* gene which encodes 1,3-D-glucan synthase. Consequently, these changes lead to cross-resistance between all echinocandin members and dramatically lower echinocandin susceptibility [111]. Additionally, naturally occurring *FKS-1* gene polymorphisms in other *Candida* spp. result in significantly greater MICs to echinocandins. For instance, in contrast to wild-type isolates of other species, such as *C. albicans* and *C. tropicalis*, *C. parapsilosis* and *C. guilliermondii* produce fundamentally higher MICs to the echinocandins [115,116]. Increased echinocandin MIC values were connected to genetic mutations in the hot spot areas of the *FKS-1* and *FKS-2* genes [117,118].

The resistant mechanisms of echinocandins in *Aspergillus* spp. are not well understood compared to those in *Candida* spp. [119]. *A. fumigatus* may develop resistance to anidulafungin. E Silva et al. demonstrated that point mutations in AF_R0_ and AF_R1_ lead to the replacement of glutamine by glutamate at position 671 of *FKS1p* (E671Q), resulting in the subsequent development of resistance in *A. fumigatus* against anidulafungin [120]. A point mutation that leads to a change in the *F675S* amino acid is responsible for the development of resistance to micafungin in patients infected with chronic pulmonary aspergillosis [121]. Moreover, a mutation that results in an *S678P* amino acid change was reported in *A. fumigatus* as well as in *Candida* isolates, and it is responsible for the development of caspofungin resistance [122,123]. Hence, the primary mechanism attributed to reduced echinocandin susceptibility in *A. fumigatus* is the point mutations in the *FKS-1* genes. However, in two clinical isolates of *A. fumigatus* resistant to caspofungin, Arendrup et al. failed to observe any mutations in the *FKS-1* gene. Instead, they demonstrated a rise in the expression of the *FKS-1* gene [124]. This mechanism might be involved in the development of tolerance to the echinocandin medication. It was demonstrated that mutations in the *FKS-1* and *FKS-2* genes, especially in the hot spot1 (HS1) region, are responsible for the development of caspofungin resistance against *C. glabrata* and other *Candida* spp. [125,126,127,128,129]. Sofia et al. discovered that the Ser663Pro substitution and Phe659 deletion cause structural changes in the HS1 of the *FKS-2* gene, which result in a significant decrease in echinocandin activity against *C. glabrata* [118].

In summary, echinocandin resistance has become a significant concern, particularly in *Candida* spp., due to mutations in the *FKS1* and *FKS2* genes, which reduce susceptibility to these drugs. This resistance mechanism limits therapeutic options, making infections more difficult to treat. Therefore, ongoing surveillance and novel treatments are crucial for managing echinocandin resistance.

### 4.3. Resistance to Amphotericin B

The development of resistance to AmB in harmful fungi rarely occurs. However, innate resistance is present in some unusual fungal pathogens, including *C. guilliermondii*, *Scedosporium apiospermum*, *A. terreus*, and some *Mucorales* spp. (such as *Cunnighamella*) [130], as well as the majority of *Fusarium* spp. [107,131].

In *Candida* spp., the development of resistance to AmB is a result of a depletion in the ergosterol level in the fungal cell membrane. Additionally, a mutational defect in the *ERG* gene, which encodes the 14-α demethylase enzyme, leads to a significant decrease in ergosterol biosynthesis. This genetic mutation may be responsible for resistance to both azoles and AmB [132]. In *C. albicans*, the depletion of ergosterol in the cell membrane causes the substitution of ergosterol with other sterol intermediates, and this also accounts for cross-resistance to azoles and AmB [133,134]. However, little is known about *Aspergillus* resistance mechanisms [107].

In summary, although amphotericin B resistance is rare, it has been observed in certain fungal species. Resistance arises through mechanisms such as reduced ergosterol biosynthesis, altered membrane composition, and stress response adaptations. These changes compromise the drug’s effectiveness, posing serious challenges for treatment.

## 5. Novel Drugs to Combat Fungal Resistance

Finding novel medications with a larger variety of functions is becoming more and more important because of the increased prevalence of fungal infections, the death rate from invasive fungal infections, and the limitations of currently available antifungal agents (Table 2). Excessive consumption, prolonged medical protocols, and environmental contamination with azoles, polyenes, and echinocandins during the previous ten years have resulted in the rapid evolution of resistance [135,136,137,138,139,140,141,142,143]. Moreover, *C. auris*’s worldwide emergence makes combating this lethal fungal infection more difficult [30]. According to data from the CDC, *C. auris* most closely resembles contagious MDR bacteria such as MRSA [144].

### 5.1. Synthetic Drugs

Luliconazole, an imidazole-type medication, was created in 2013 and is available under the brand name Luzu. It has FDA approval for interdactylic tinea pedis, tinea corporis, and tinea croris, which are all caused by fungi [145]. Luliconazole’s molecular structure makes it less likely to attach to keratin and makes it easier to penetrate the nail layer. In vitro tests have demonstrated that luliconazole is effective against *A. fumigatus*, *C. albicans*, *Malassezia* sub spp., and *Trichophyton rubrum* [146].

Cresemba is an azole antifungal that disrupts ergosterol synthesis in fungal cell membranes. On 6 March 2015, the US FDA authorized cresemba (isavuconazonium sulfate), as a novel antimicrobial medication for the management of invasive aspergillosis and invasive mucormycosis in adults [147,148].

VT-1161, VT-1129, and VT-1598 are novel tetrazole medications with improved selectivity for fungi. All of them have the effect of suppressing the fungus *CYP51*, with an affinity that is more than 2000 times greater than that of comparable enzyme sites in humans [149]. VT-1161 (osteoconazole) is a novel, oral drug and its selected experimental trials are now in advanced clinical development. It was reported that *C. glabrata* and *C. krusei* isolates resistant to azoles and echinocandins are susceptible to the in vitro activity of VT-1161 [150]. Moreover, in a new study, Stephen et al. reported the high efficacy of VT-1161 in managing acute vulvovaginal candidiasis [151]. The FDA has designated VT-1129 as an orphan drug. The structure of the molecule is extremely close to VT-1161. Also, it shows promising effects against *C. glabrata* and *C. krusei* [152]. Nowadays, VT-1129 is being studied in phase I clinical trials and is primarily used to combat cryptococcal meningitis [153]. Additionally, VT-1598 is another novel tetrazole analog and was demonstrated to have in vitro and in vivo activity against *C. auris* [154]. Also, it shows good activity in combating invasive aspergillosis [155] and cryptococcal meningitis in an animal model [156].

T-2307 is a novel class of arylpyrimidine derivatives with an action mechanism distinct from that of existing antifungal medications now available. It interferes with the production of energy in the cell process in fungi by rupturing the mitochondrial membrane [157]. T-2307 is absorbed by fungal cells via high-affinity polyamine transporters, where it inhibits respiratory chain complexes III and IV, disrupting mitochondrial membrane potential [157]. In a previous study, T-2307 can selectively inhibit mitochondrial activity in yeasts [158]. Also, it exhibits promising in vitro and in vivo activity against *C. auris* [159], as well as some other *Candida* spp. [160,161] and *Cryptococcus gattii* [162].

Rezafungin (CD101) is a novel β-glucan synthase inhibitor that is chemically related to anidulafungin but has improved tissue penetration [163], pharmacokinetic/pharmacodynamic (PK/PD) pharmacometrics [164,165], and stability. Other echinocandins and rezafungin share the same targets and mechanism of action, but rezafungin has a better safety profile and may be administered at greater dosages. Eventually, higher dosage regimens will eliminate (or lessen) the selection of resistant strains [166]. Compared to previous echinocandins, it is far more stable in solution, giving it greater dosage, storage, and production flexibility. These characteristics would enable the development of novel administration methods, such as topical and subcutaneous applications [167], and would enable rezafungin to be delivered once weekly (intravenously) [168]. It has extremely strong in vitro anti-*Aspergillus* spp. action (including MDR cryptic species) [169,170]. However, its spectrum is limited. Rezafungin is inherently resistant to *Basidiomycetes*, *Mucorales*, *Fusarium* spp., and *Ajellomycetaceae* [171].

### 5.2. Natural Drugs

Ibrexafungerp (IBX, SCY-078), a derivative of a triterpene natural substance, is a promising lead in Phase III clinical trials. Although ibrexafungerp inhibits 1,3-b-D-glucan synthase as echinocandins, it has a unique structure and binds to a different location on the enzyme, hence resistance mechanisms linked to echinocandins do not influence ibrexafungerp’s activity [172]. In recent years, ibrexafungerp was exclusively studied for its ability to combat vulvovaginal candidiasis [173,174,175,176].

Nikkomycin Z is another example of a novel natural drug, and it is now being tested in Phase II clinical trials. It is a competitive inhibitor of chitin synthase; so, it prevents antifungal activity by inhibiting the synthesis of the fungal cell wall [177]. Nikkomycin Z has been reported to combat coccidioidomycoses [178]. Furthermore, it has the ability to eradicate most *C. auris* isolates [179]. Additionally, it shows synergistic activity against *C. albicans* and *C. parapsilosis* biofilms when combined with caspofungin and micafungin [180].

The fungus siderophore VL-2397 is another Phase II clinical medication being investigated for the management of invasive aspergillosis infections [181]. It is produced by the fungal strain MF-347833, which was isolated from Malaysian leaf litter [182]. Siderophore iron transporter 1 (Sit1) seems to be the pathway by which VL-2397 enters fungal cells [181].

The pre-clinical drug candidate aureobasidin A is a fungal cyclic depsipeptide with action against *Candida*, *Aspergillus*, and *Cryptococcus* spp. Through a novel target that is unique to fungi, aureobasidin A prevents cell membrane synthesis. It is the first agent that prevents inositol phosophorylceramide (IPC) synthase from producing sphingolipids [183,184].

## 6. New Approaches and Strategies to Combat Fungal Resistance

The lack of viable alternative tactics for treating fungal infections is caused by the limited antifungals and well-documented resistance against the antifungal drugs that are now accessible. Since fungi are eukaryotic, antifungal medications may pose risks to the host due to their potential to affect human cells.

### 6.1. Combinatorial Therapy

Combinatorial therapy enhances treatment effectiveness and reduces drug resistance more effectively than monotherapy [185,186,187]. It works by targeting multiple biological pathways for better efficacy and bioavailability. This approach includes using two antifungal drugs or pairing an antifungal with a non-antifungal to boost its effectiveness, for instance, with calcineurin (as cycloserine), lysine deacetylase, lysine acetyltransferase (as MGCD290), or Hsp90 inhibitors [188].

### 6.2. Drug Repurposing

Drug repurposing offers a cost-effective and time-saving alternative for developing new antifungal drugs. This approach involves using computer modeling and molecular docking to identify existing drugs with antifungal potential, followed by laboratory experiments to confirm their effectiveness [189] (Figure 4). Based on this methodology, recent investigations have found possible antifungal activity with well-known medications like anti-inflammatory drugs (such as ibuprofen and aspirin), anti-rheumatic drugs (such as auranofin), anti-cancer drugs (such as tamoxifen), lipid-lowering drugs (such as atorvastatin), and Ca^2+^ channel blockers (such as felodipine and nifedipine) [190]. Other drugs include cisplatin, raltegravir, pitavastatin, aripiprazole, benzimidazoles, and quinacrine. They all provided therapeutic advantages in infected animal models and were proven to reduce hyphal growth and biofilm development [189].

Wiederhold et al. [191] proved the antifungal activity of auranofin and documented its effectiveness against *C. albicans* biofilms. Also, auranofin was reported to have activity against *Cryptococcus* spp. that cause serious infection in an immunodeficient patient. Additionally, another study demonstrated the relationship between tamoxifen and its antifungal activity and concluded that it is due to the estrogen receptor-independent mechanism of tamoxifen [192]. Several anti-inflammatory medications, including aspirin, ibuprofen, and tacrolimus, have demonstrated antifungal activity against *C. neoformans*, *C. gattii*, and *E. rostratum*, respectively [193]. Furthermore, some Ca^2+^ channel blocker drugs showed activity against different fungal species such as *Cryptococcus*, *Candida*, *Saccharomyces*, and *Aspergillus* [194]. Atorvastatin was evaluated as an auxiliary agent to treat fungal infections in a trial that demonstrated the efficacy against one strain of *C. gattii* [195].

### 6.3. Immuno-Modulation

Enhancing the immune system is a promising strategy to fight invasive fungal infections. Studies in animal models show that adding immune-stimulating factors, such as G-CSF and GM-CSF, improves antifungal drug effectiveness against *Candida*, *Aspergillus*, and *Cryptococcus* spp. [196,197] Additionally, INF-γ improves the response to antifungal medications in invasive aspergillosis animal models. Moreover, monoclonal antibodies with specificity for the fungal cell surface improve the survival of *Candida*-, *Aspergillus*-, and *Cryptococcus*-infected animal models [196]. Also, the transfusion of granulocytes, T cells, and natural killer (NK) cells was shown to improve longevity in animal models of invasive fungal infections [197]. Additionally, live attenuated fungal cells or cell wall parts have been shown to activate adaptive immune responses and provide prophylaxis from fungal infections [196].

### 6.4. Use of Probiotics

Probiotics were classified by the WHO in 2002 as live organisms that, when administered in sufficient amounts, are helpful to the health of humans [198].

Probiotics prevent fungal infections by inhibiting growth, adhesion, and biofilm formation. Although the exact mechanism is unclear, studies suggest that probiotics interact with harmful fungi by competing for nutrients and receptor sites, limiting fungal survival [199,200,201]. For instance, probiotics and pathogens can compete for nutrients, growth factors, and receptor binding sites when they are cultured with each other [202,203]. Moreover, the production of secondary metabolites such as lactic acid, acetic acid, and hydrogen peroxide assesses probiotics’ role in combating harmful fungi [204,205,206].

### 6.5. Antifungal Lock Therapy

*Candida* spp. can stick to indwelling device surfaces, such as catheters, and build a substantial extracellular polysaccharide matrix that will assist in the formation of biofilms [207]. According to recent recommendations, if a catheter infection caused by *Candida* spp. is confirmed, it must be replaced immediately [208,209]. However, changing central venous catheters (CVCs) can be challenging, and depending on where the catheter is inserted, the exchange greatly raises the risk of complications and bloodstream infections [209]. The term lock therapy describes the process of pumping a concentrated antibiotic solution into the catheter lumen and letting it stay there for a certain amount of time in order to maintain a sustained antibiotic level that is sufficient to kill bacteria embedded in the catheter’s biofilm [24]. The use of antibiotic lock techniques has been suggested for the prevention and treatment of several catheter-associated infections since the late 1980s. Till now, there was no formally approved antifungal lock treatment approach. However, the majority of bloodstream infections caused by *candida* spp., as a result of intraluminal colonization, form a continuous source of life-threatening candidemia. According to the in vitro, in vivo, and clinical data that are currently available, ethanol-based lock solutions exhibit the maximum activity [210]. According to in vitro data published by Alonso et al., a lock solution containing 40% ethanol and 60 IU heparin, applied daily for 72 h, was nearly effective in eliminating the metabolic activity of biofilms formed by a *C. albicans* reference strain [211]. However, the hazards and restrictions associated with ethanol prevent its broad clinical usage [212]. Therefore, alternative lock solutions have been explored, including taurolidine–citrate lock solutions [213]. Several clinical trials have investigated the efficacy of taurolidine–citrate lock solutions, which exhibit both antimicrobial and anti-inflammatory properties, reducing biofilm formation without the cytotoxic effects associated with ethanol [214,215].

The combination of echinocandin and AmB-based lock solution, either alone or in conjunction with prospective adjunctive drugs with different mechanisms, has been studied in vitro; however, few reliable in vivo studies address this combination [24,216]. A recent study showed that aspirin might be used as a lock component [217].

### 6.6. Antifungal Enzymes

The human body spontaneously produces and secretes lysozymes to protect against harmful bacteria and fungi [218]. It has been discovered that five antimicrobial peptides produced from lysozyme during pepsin digestion are effective against *C. albicans* [219].

#### 6.6.1. Chitotriosidase

Activated human neutrophils and macrophages secrete the enzyme chitotriosidase, which is known to split chitin and eradicate invasive mycoses. It was reported that the chitotriosidase enzyme is present in lacrimal secretion and expressed in large amounts as a defense mechanism against harmful fungal pathogens [220]. Currently, chitotriosidase is employed as a marker of macrophage activation brought on by conditions like sarcoidosis and different lipid storage conditions, such as Gaucher disease (GD). Animal models have shown upregulation in the expression of chitotriosidase mRNA, indicating that they are crucial mediators of immunological responses [221].

#### 6.6.2. Lactoferrins

Salivary secretions and nearly all other bodily fluids contain lactoferrins, which are proteases that bind to iron atoms. Human-produced lactoferrins have been reported to have anti-*C. krusei* and anti-*C. albicans* activity [222,223]. Clinical trials showed that orally ingested bovine lactoferrins can dramatically lessen already progressed candidiasis in mouse models. Also, lesions in the oral cavity decreased because of the therapeutic impact of lactoferrins [224,225].

#### 6.6.3. Antileukoproteases

Antileukoproteases, which are mucous protease inhibitors based on serine, are found in body secretions from the mucosa of the bronchial, cervical, and nasal passages as well as saliva and seminal fluids. Tomee et al. [226] investigated the antifungal activity of recombinant antileukoproteases against *A. fumigatus* and *C. albicans*. Antileukoproteases have been advised to be a method of treatment for fungal infections in immunodeficient people as well as a substitute for conventional antifungal medications [226].

### 6.7. Antimicrobial Peptides

AMPs, or antimicrobial peptides, are potent antibacterial, antifungal, and antiviral agents. All organisms produce AMPs, which make natural and synthetic AMPs an important source for treating infections. AMPs are interesting potential agents for treating infections because of their strong efficacy at inactivating pathogens and sometimes simultaneous immunomodulatory properties [227]. Numerous antifungal peptides with antimicrobial and immunomodulatory properties have been found. The body of a human also secretes a lot of these peptides as a defense mechanism against potential pathogens and invasive mycoses. Jelleine-1, a promising peptide, has been demonstrated to possess extremely powerful in vivo and in vitro antifungal properties [228].

#### 6.7.1. The Human GAPDH Peptide (hGAPDH)

It has been demonstrated that a peptide generated from the conserved protein, GAPDH, offers some sort of tissue defense against fungal infections. The human GAPDH peptide, or hGAPDH, has been demonstrated to simultaneously suppress virulence factors and the multiplication of *C. albicans* at low concentrations [229].

#### 6.7.2. Defensins

Defensins are a class of short, positively charged peptides with cysteine residues that may be distinguished by the orientation of their disulfide bonds. It is known that the human body can synthesize defensins (α and β defensins.) Defensins have been discovered in a broad range of eukaryotes and prokaryotes and are not just found in humans [230]. Krishnakumari et al. [231] studied the effect of different types of human defensins along with minimal fungicidal concentrations and reported significant fungicidal efficacy against *C. albicans* [231]. Defensins derived from plants and insects are also effective in combating harmful fungi. For example, it has been demonstrated that the defensins “HsAFP1” from plants and “Heliomicin” from insects are particularly efficient in eradicating *C. albicans* [232].

#### 6.7.3. Dermcidin

Dermcidin is a natural peptide synthesized in the human sweat gland and then secreted on the skin surface. Dermcidin has been evaluated in vitro and was found to have strong antifungal activity against *C. albicans* in the presence of artificial conditions that resemble sweat components [233]. Also, it was reported that the expression of dermcidin was downregulated in tinea pedis patients [234].

#### 6.7.4. Cathelicidin

Cathelicidin is a polypeptide that is accumulated in high amounts in macrophages and polymorphonuclear leukocytes. Their main function is to protect exposed body surfaces like the skin from infections and invasions by acting as an antibacterial defensive barrier. This peptide significantly influenced the growth and metabolic activity of several medically important fungal spp., including some azole-resistant isolates such as *A. fumigatus* [235]. In a different investigation, five distinct cathelicidins showed substantial in vitro action against clinically isolated yeasts, particularly *C. neoformans* [236].

#### 6.7.5. Synthetic Peptides

Mo-CBP3-PepI and Mo-CBP3-PepIII are two examples of novel antifungal peptides. Leandro et al. reported their synergistic activity when co-administered with conventional antifungals, such as nystatin and itraconazole, against *C. albicans* and *C. parapsilosis* [237]. In another study, synthetic peptides Mo-CBP3-PepI, Mo-CBP3-PepII, Mo-CBP3-PepIII, RcAlb-PepI, RcAlb-PepII, RcAlb-PepIII, PepGAT, and PepKAA were shown to have promising activity against *C. neoformans* and exert several mechanisms for membrane pore formation, DNA degradation, and apoptosis [238]. In the most recent study, new classes of dipeptide, tripeptide, Trp-His(1-Bn)-OMe/NHBn, and His(1-Bn)-Trp-His(1-Bn)-OMe/NHBn were evaluated, and they showed promising results against *C. neoformans* through the lysis of the fungal cell wall [239].

### 6.8. Nanoparticles (NPs)

Nanoparticles offer multiple benefits in antifungal therapy. They extend drug release, protect drugs from breakdown, improve therapeutic effectiveness, and help prevent the development of drug resistance [30,240,241,242,243,244,245,246,247,248,249,250]. The difficulties in treating fungi can be overcome by using metallic NPs, mesoporous silica NPs, polymeric NPs, and lipid-based nanosystems [251].

#### 6.8.1. Metallic NPs

Metal NPs range in size from 1 to 100 nm. They have several benefits such as high stability, significant antifungal properties, minimal cytotoxicity, and poor pathogen resistance [245,246,252,253,254,255]. Also, they possess antifungal activity through their ability to prevent the formation of specific fungal proteins and cell membranes in addition to promoting the generation of fungal ROS [256,257,258].

It has been demonstrated that gold NPs have anti-*candida* properties with low toxicity. To increase the antifungal effects of gold NPs, effective drugs are conjugated with them. For instance, indolicidin, a host defense peptide, was coupled with gold nanoparticles to treat *C. albicans* that showed resistance to fluconazole [259]. In a different investigation, the aspartyl proteinase 2 (Sap2) that is released by *C. albicans* was inhibited using triangular gold NPs coupled with certain peptide ligands [260].

It has been demonstrated that silver NPs have significant potential for preventing resistance in bacteria and inhibiting the growth of fungi (Figure 5). Studies conducted in vitro using silver NP monotherapy found that *C. albicans* and *C. tropicalis* growth and survival were considerably inhibited [261,262]. Silver and gold NPs have also been bio-synthesized to fight skin infections caused by fungi. Silver NPs could reduce the growth of *Candida*, *Microsporum*, and *Trichophyton* dermatophyte isolates, although *C. neoformans* was susceptible to both gold and silver nanoparticles [263]. Most recently, Bharti et al. [264] synthesized bio-stabilized silver NPs from lichens and discovered that their nano-system can efficiently eradicate fluconazole-resistant *C. albicans*.

Additionally, zinc and zinc oxide NPs are frequently researched. Zinc NPs have been shown to provide significant anti-candida advantages when used alone or in combination with antifungal drugs such as fluconazole [265,266].

#### 6.8.2. Mesoporous Silica NPs

Mesoporous silica NPs (MSNs) are promising in the field of drug delivery due to their great chemical and thermal stability, biocompatibility, and wide surface area for delivering therapeutic carriers [267]. It has been proven that adding econazole to the MSN system provides antifungal action toward *C. albicans* [268]. Additionally, MSNs have been combined with metallic NPs to augment the fungicidal properties. Furthermore, the eugenol-loaded MSN system was demonstrated, through in vitro tests, to decrease the proliferation of *A. niger*. Tebuconazole was delivered via a pH-sensitive, controlled MSN system, so this formed nanosystem dramatically reduced yeast growth, and it may be recommended for the treatment of vaginal candidiasis [269].

#### 6.8.3. Polymeric NPs

Polymeric nanoparticles are between 5 and 1000 nm in size. Antifungal medications and other active substances can be placed onto polymeric cores or trapped inside particles. Treatments based on polymeric NPs provide advantages such as increased therapeutic effectiveness and penetration, decreased toxicity, and pharmacological action delivered to a specific target [270,271,272]. Chitosan was integrated into polymeric NPs in cases of oral candidiasis, which prevented *Candida* from growing and forming a biofilm [273]. Also, Costa et al. [274] loaded chitosan NPs with farnesol, a natural essential oil with antimicrobial properties, and found that this nanosystem can reduce the fungal load of *C. albicans* and inhibit biofilms and hyphae. Additionally, an efficient therapy for cryptococcal meningitis was produced by loading AmB onto polybutylcyanoacrylate NPs, which have a high blood–brain barrier (BBB) crossing ability [275]. Li et al. [276] created a novel drug carrier using chitosan-conjugated poly (lactic-co-glycolic acid, PLGA) NPs to treat Cryptococcal-related lung infections.

#### 6.8.4. Liposome NPs

Liposomes were initially created as double-layer lipid delivery methods for drugs. Phospholipids, which have both hydrophobic and hydrophilic properties, make up the majority of the liposomal structure’s chemical constituents. Drugs with different polarities can be incorporated into liposomal structures thanks to these components: hydrophilic compounds are contained in the aqueous zone, while hydrophobic chemicals are retained inside the liposome [277]. To avoid the harmful side effects and nephrotoxicity of AmB, LAmB conjugates were created as an alternative to AmB deoxycholate injections [251,278]. LAmB was the first FDA-approved and marketed liposomal formulation for fungal infections [279]. Several clinical trials were conducted to evaluate the spectrum of activity and efficacy of LAmB. LAmB showed promising effects against several fungal infections such as mucormycosis [280,281,282], aspergillosis [283,284,285], candidiasis [286,287,288], *Candida* meningoencephalitis [289], cryptococcosis [88,290], histoplasmosis [291], and other uncommon but highly invasive mycoses, such as fusariosis [292].

Vera-González et al. fabricated anidulafungin liposomes for the first time and showed that these liposomes have MICs comparable to free anidulafungin for suppressing planktonic *C. albicans* growth. Also, they demonstrated that these liposome formulations can penetrate *C. albicans* biofilms more effectively than free anidulafungin [72].

Recently, Paosupap et al. [293] discovered that nanoliposomes containing rhinacanthin-C (natural antifungal extract) can increase the stability and antifungal activity of rhinacanthin-C, with sustained and prolonged durations of activity.

In summary, while nanotechnology and drug repurposing offer promising antifungal strategies, they should be considered alongside combinatorial therapy, antifungal lock therapy, and antimicrobial peptides. Nanotechnology provides enhanced drug delivery but faces regulatory and toxicity concerns. Drug repurposing accelerates antifungal discovery but has variable efficacy. Combinatorial therapy prevents resistance but increases drug interactions. Antifungal lock therapy remains limited to catheter-associated infections. Antimicrobial peptides represent a novel approach with potent antifungal activity and lower resistance risks but require further optimization for stability and clinical application. An integrated, multi-faceted approach is essential to effectively combat antifungal resistance.

In Table 3, we critically summarize and compare the new approaches to combat fungal resistance in terms of mechanism, efficacy, limitations, and clinical applicability.

## 7. Conclusions

As fungal resistance continues to rise, there is a growing need for new antifungal therapeutic options, particularly in cases involving rare fungal species or resistance to multiple drugs. This review has explored the underlying mechanisms of antifungal resistance and treatment failure, examined conventional therapies along with their mechanisms of action and resistance pathways, and highlighted emerging antifungal drugs and novel therapeutic strategies.

Conventional antifungal treatments, including azoles, echinocandins, and polyenes, remain the cornerstone of antifungal therapy but are often limited by toxicity, resistance development, and spectrum restrictions. In contrast, novel therapeutic approaches, such as combinatorial therapy, drug repurposing, and nanotechnology-based drug delivery, offer promising alternatives with enhanced efficacy, reduced toxicity, and improved resistance management. These emerging treatments represent a shift toward more targeted and effective antifungal strategies. Future research should prioritize novel antifungal discovery, optimized combination therapies, and advanced drug delivery systems for improved stability and targeting. Additionally, host-directed therapies aimed at strengthening the immune response and reducing fungal virulence hold significant promise.

Addressing translational challenges, such as ensuring safety, efficacy, and scalability, will be critical for the successfully integration of these advancements into clinical practice. The continuous evolution of antifungal resistance necessitates sustained research efforts to refine therapeutic strategies and improve patient outcomes in the fight against invasive fungal infections.

## 8. Future Directions and Final Remarks

Antifungal therapy is advancing with promising new approaches to overcome current treatment limitations. Research should prioritize the development of safer, more effective antifungal agents while exploring novel drug delivery systems that enhance bioavailability and minimize side effects. Combining antifungal therapies with host-targeted strategies may enhance patient outcomes by reducing fungal virulence and resistance. Further studies are also needed to evaluate the clinical feasibility of emerging treatments such as nanoparticle-based drug formulations and immunomodulatory therapies. By focusing on these avenues, future antifungal strategies can provide more effective and sustainable solutions for combating fungal resistance, ultimately improving patient care and public health outcomes.

## Figures and Tables

**Figure 1 pharmaceuticals-18-00364-f001:**
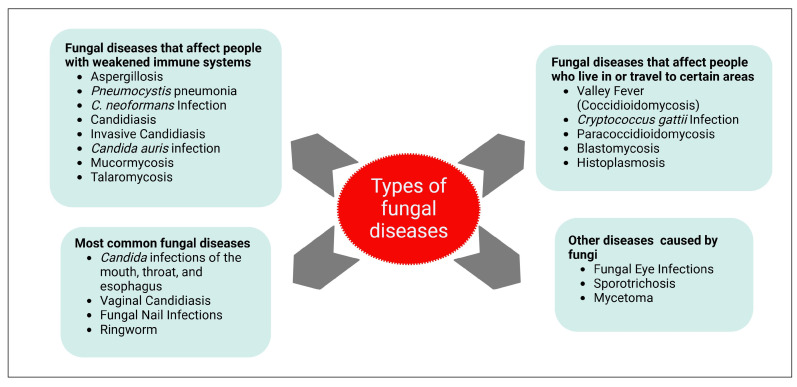
Types of fungal diseases according to CDC categorization. CDC categorizes the illnesses brought on by fungi into four groups: the most prevalent fungal illnesses, those that affect people who live in or travel to particular regions, those that affect immunocompromised people, and other illnesses and health issues brought on by fungi. Created with BioRender.com.

**Figure 2 pharmaceuticals-18-00364-f002:**
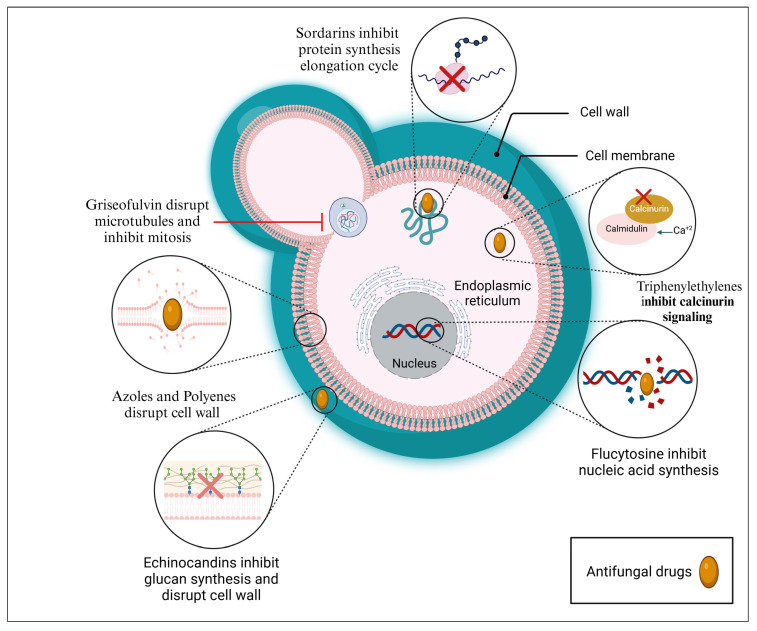
Mechanisms of action of antifungal drugs. They act through the disruption of the cell wall, the disruption of the cell membrane, the inhibition of protein synthesis, the inhibition of nucleic acid synthesis, the disruption of microtubules, or the inhibition of calcinurin signaling. Created with BioRender.com.

**Figure 3 pharmaceuticals-18-00364-f003:**
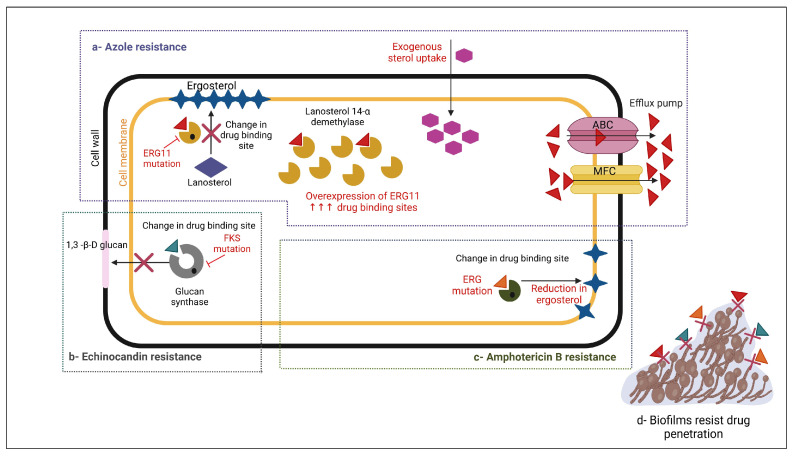
Mechanism of resistance to common antifungal medications. (**a**) Azole resistance develops as a result of point mutations or the overexpression of the *ERG11* gene, which encodes the enzyme lanosterol-14-demethylase. (**b**) Echinocandin resistance develops through a mutation or substitution in the *FKS-1* gene, which encodes the β-1,3 glucan synthase enzyme. (**c**) Polyene resistance develops as a result of the ERG mutation, which leads to decreased ergosterol biosynthesis. (**d**) Some fungi can form biofilms and acquire resistance to nearly all antifungal classes. Created with BioRender.com.

**Figure 4 pharmaceuticals-18-00364-f004:**
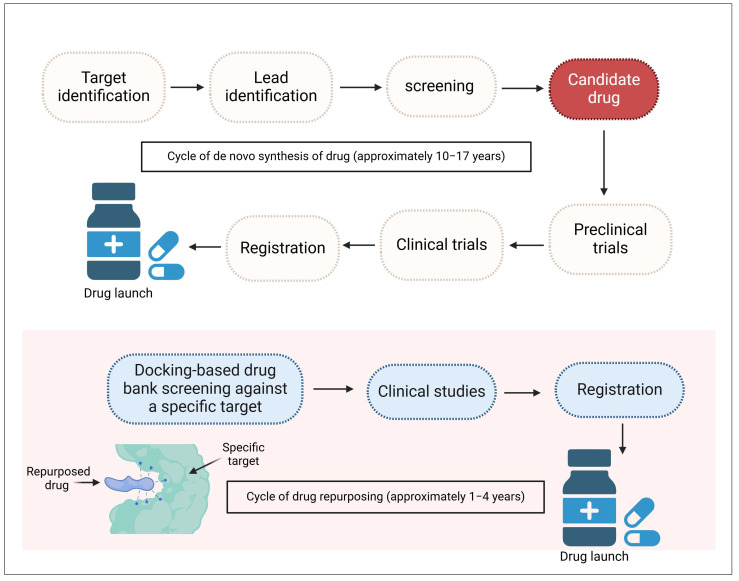
Difference between de novo synthesis and the repurposing of a drug. The repurposing drug approach overcomes high costs, long periods, and efforts to develop novel drugs. Created with BioRender.com.

**Figure 5 pharmaceuticals-18-00364-f005:**
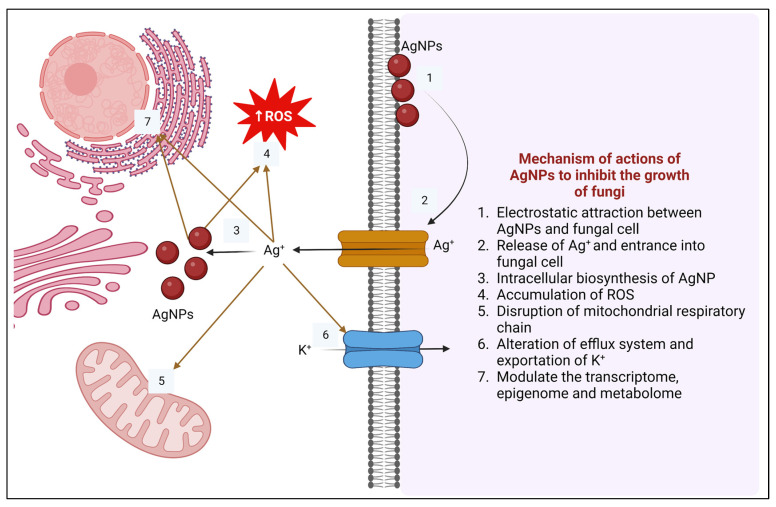
Mechanism of action of AgNPs to combat fungal resistance. Created with BioRender.com.

**Table 1 pharmaceuticals-18-00364-t001:** A summary table comparing antifungal drugs, their mechanisms of action, and clinical applications.

Drug Class	Example Drugs	Mechanism of Action	Clinical Applications
Azoles	Fluconazole, Itraconazole, and Voriconazole	Inhibit ergosterol synthesis by targeting lanosterol 14α-demethylase, leading to membrane instability.	Candidiasis, aspergillosis and cryptococcosis
Echinocandins	Caspofungin, Micafungin, and Anidulafungin	Inhibit β-glucan synthase, disrupting fungal cell wall integrity.	Invasive candidiasis and aspergillosis
Polyenes	Amphotericin B and Nystatin	Bind to ergosterol, increasing membrane permeability and causing cell death.	Systemic fungal infections and oral and vaginal candidiasis
Allylamines	Terbinafine and Naftifine	Inhibit squalene epoxidase, blocking ergosterol synthesis.	Dermatophytoses (e.g., ringworm and athlete’s foot)
Pyrimidine Analogs	Flucytosine	Converted to 5-fluorouracil, disrupting fungal RNA and protein synthesis.	Cryptococcal meningitis and candidiasis (used in combination therapy)
Griseofulvin	Griseofulvin	Disrupts microtubule function, inhibiting mitosis.	Dermatophytoses (e.g., tinea infections)

**Table 2 pharmaceuticals-18-00364-t002:** The efficacy, side effects, and resistance potential of synthetic and natural antifungal drugs.

Drug Name	Type (Synthetic/Natural)	Mechanism of Action	Efficacy	Side Effects	Resistance Potential
Luliconazole	Synthetic	Inhibits ergosterol synthesis	High	Mild skin irritation	Low
Cresemba (Isavuconazonium sulfate)	Synthetic	Inhibits ergosterol synthesis	High	Mild liver toxicity and nausea	Low
VT-1161, VT-1129, and VT-1598	Synthetic	Inhibits fungal CYP51	High	Minimal side effects	Low
T-2307	Synthetic	Disrupts mitochondrial function	High	Potential mitochondrial toxicity	Low
Rezafungin (CD101)	Synthetic	Inhibits β-glucan synthesis	High	Gastrointestinal discomfort	Low
Ibrexafungerp	Natural	Inhibits β-glucan synthesis	High	Gastrointestinal discomfort	Low
Nikkomycin Z	Natural	Inhibits chitin synthesis	Moderate	Minimal side effects	Low
VL-2397	Natural	Targets iron transport in fungi	High	Limited data	Unknown
Aureobasidin A	Natural	Inhibits IPC synthase	High	Limited data	Unknown

**Table 3 pharmaceuticals-18-00364-t003:** Summary of new approaches to combat fungal resistance.

Strategy	Mechanism	Advantages	Limitations	Key Studies
Nanotechnology	Enhances drug delivery via nanoparticles (e.g., liposomes and metallic NPs)	Improves bioavailability, reduces toxicity, and overcomes resistance	High cost, regulatory hurdles, and potential toxicity	[72,261,262,264,268]
Drug Repurposing	Uses FDA-approved drugs with antifungal potential	Fast-track approval, low cost, and established safety	Variable efficacy and risk of resistance	[191,192,193,194,195]
Combinatorial Therapy	Uses multiple antifungal or antifungal + non-antifungal agents	Synergistic effects and delays resistance	Drug–drug interactions and worse side effects	[188]
Antifungal Lock Therapy	Direct antifungal application in catheters to prevent bloodstream infections	Targets biofilms and localized effects	Limited to catheter-related infections	[210,212]
Antimicrobial Peptides (AMPs)	Naturally occurring or synthetic peptides that disrupt fungal membranes and modulate immune response	Broad-spectrum activity, low resistance potential, and synergy with antifungals	Stability issues, susceptibility to degradation, and limited clinical trials	[229,232,233,236,239]

## Data Availability

Not applicable.

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
