# Peer review of "Beyond Conventional Antifungals: Combating Resistance Through Novel Therapeutic Pathways"

_pharmaceuticals, 2025, doi:10.3390/ph18030364_

Round 1
Reviewer 1 Report
Comments and Suggestions for Authors
Publication reviews are very important for the rapid development of science. In this manuscript, the authors briefly review the traditional mechanisms of drug action and the target of new drugs to treat fungal resistance. In addition, they review recent research on new approaches to prevent fungal resistance and treat fungal infections that are resistant to multiple drugs. This review focuses on new and existing drug targets, as well as potential mechanisms for new drug development. A review of information on potential targets could help scientists more quickly find new potential compounds to treat multidrug-resistant fungal infections.
In order to improve this review, I would like the manuscript to be supplemented with the chemical structures of the compounds it discusses. Also, the resolution of the images has been improved. Self-citation should be reduced to at least 5%, now I have found 8%.
Reduce general information by eliminating citations of sources from non-scientific articles.
Organize the reference section, unify the list and organize it according to the standard.
I would recommend to consider reorganizing small subsections because important information is not provided.
Author Response
Reviewer 2
Publication reviews are very important for the rapid development of science. In this manuscript, the authors briefly review the traditional mechanisms of drug action and the target of new drugs to treat fungal resistance. In addition, they review recent research on new approaches to prevent fungal resistance and treat fungal infections that are resistant to multiple drugs. This review focuses on new and existing drug targets, as well as potential mechanisms for new drug development. A review of information on potential targets could help scientists more quickly find new potential compounds to treat multidrug-resistant fungal infections.
In order to improve this review, I would like the manuscript to be supplemented with the chemical structures of the compounds it discusses.
Response: Thanks for the reviewer for this suggestion. We actually not specialists in chemistry and are not qualified to provide chemical structure. Also, we focus here on the mechanisms of resistance and how to combat.
Also, the resolution of the images has been improved.
Response: We appreciate the reviewer’s feedback regarding the figures. All revised figures have been updated in the manuscript with high resolution.
Self-citation should be reduced to at least 5%, now I have found 8%.
Response: Thank you for your insightful comment. I have reviewed the manuscript and reduced the self-citations to align with the recommended threshold of 5%.
Reduce general information by eliminating citations of sources from non-scientific articles.
Response: Thank you for your feedback. I understand the concern about the use of non-scientific sources. We focused on retaining references from peer-reviewed sources to ensure the information presented is more aligned with the scientific nature of the paper.
Organize the reference section, unify the list and organize it according to the standard.
Response: Thank you for pointing that out. The references are now unified and formatted according to the standard style required for the manuscript.
I would recommend to consider reorganizing small subsections because important information is not provided.
Response: Thank you for your valuable feedback. I have carefully reviewed the structure of the small subsections and reorganized them to improve the logical flow of information. Additionally, I have ensured that all key points are clearly presented to avoid any gaps in important details.

Reviewer 2 Report
Comments and Suggestions for Authors
This review highlights the growing challenge posed by fungal infections, which often go undetected in early stages, leading to severe disease progression and treatment difficulties. It discusses the limited antifungal drug classes and the rising issue of drug resistance, which significantly impacts patient management. The review provides an overview of conventional antifungal drugs, their mechanisms of action, and resistance development while also exploring newly approved treatments and those in clinical trials. Additionally, it examines innovative strategies being researched to combat fungal resistance, including multi-drug resistance. Here the following major and minor comments below:
Title: The title appears exaggerated and could benefit from a more scientifically objective tone. The overuse of buzzwords such as "Revolutionizing," "Cutting-Edge," and "Game-Changing" reduces its clarity and impact. Consider rewording it for precision and conciseness.
Abstract: The abstract is wordy, lacks clarity, and would benefit from a stronger conclusion. There is redundancy in phrasing, and a more structured flow with precise language would enhance readability. A final sentence reinforcing the significance of novel strategies and future research directions in antifungal treatment is recommended for better impact.
Graphical Abstract: The quality of the graphical abstract is poor, with low pixel resolution making it difficult to interpret. A clearer, high-resolution version should be provided to enhance readability and comprehension.
Section 1: Introduction
-
The introduction should first provide an overview of fungal biology, followed by their advantages and disadvantages before discussing fungal infections.
-
The application of fungi should be mentioned, and references should be cited, such as:
-
https://doi.org/10.1093/g3journal/jkac224
-
https://link.springer.com/article/10.1007/s11274-023-03737-7
-
-
The transition to the research gap needs to be strengthened to highlight the importance of the study.
Section 2:
-
Some content is redundant, particularly in discussions on immunosuppression and invasive procedures. Streamlining these sections would enhance readability.
-
Some sentences are overly complex. Simplifying them into shorter sentences would make the text easier to follow.
Figures 1 & 2:
-
The graphical quality of Figures 1 and 2 is poor. Please improve the resolution and clarity.
-
In Figure 2, the fungus is depicted as yeast budding. However, fungi have various morphological forms that infect humans. Consider including multiple fungal morphologies.
Section 3:
-
A summary table comparing antifungal drugs, their mechanisms of action, and clinical applications would improve readability and comprehension.
-
Some statements require more recent references, particularly those discussing resistance mechanisms and emerging antifungal treatments.
Section 4:
-
This section provides good coverage of resistance mechanisms; however, a brief summary at the end of each subsection would reinforce key takeaways.
Section 5:
-
The manuscript lists various synthetic and natural antifungal drugs but lacks a comparative analysis. A table comparing their efficacy, side effects, and resistance potential would be beneficial.
Section 6:
-
Some sentences are unnecessarily complex. Simplifying these sentences by breaking them into smaller parts will improve readability.
-
The manuscript inconsistently refers to fungal species, alternating between full names (e.g., Candida albicans) and abbreviations (C. albicans). Standardizing the naming format would enhance clarity.
-
The discussion on antifungal lock therapy (Lines 484–505) mentions ethanol-based solutions but lacks alternative approaches. Including ongoing clinical trials or alternative solutions would strengthen this section.
-
Minor grammatical errors and awkward phrasing are present. For example, "antifungal medications can be hazardous to the host because fungi are eukaryotic" (Line 415) could be reworded for better clarity.
New Section Recommendation: Enzymes Involved in Antifungal Resistance
-
A new section should be added discussing enzymes that contribute to antifungal resistance.
-
The section should cover key enzymes, their role in resistance mechanisms, and potential therapeutic targets.
-
The following references should be cited:
-
http://doi.org/10.1155/2017/9870679
-
https://doi.org/10.1016/bs.mie.2023.12.016
-
Section 7: Conclusion
-
The conclusion provides a comprehensive overview of antifungal resistance but should include a structured comparison between novel and conventional treatments in terms of efficacy and safety.
-
A dedicated section on future research needs and emerging therapeutic trends would significantly enhance the review.
-
Potential areas for further exploration, such as alternative drug delivery systems, combination therapies, and host-directed strategies, should be highlighted.
-
A final section titled 'Future Directions and Final Remarks' should be included to summarize key takeaways and outline promising research avenues in antifungal therapy.
The section's English could be clearer and more concise to better convey the research. Sentences are often long and complex, making readability difficult. Improved structure, reduced redundancy, and stronger transitions would enhance clarity.
Author Response
Reviewer 3
This review highlights the growing challenge posed by fungal infections, which often go undetected in early stages, leading to severe disease progression and treatment difficulties. It discusses the limited antifungal drug classes and the rising issue of drug resistance, which significantly impacts patient management. The review provides an overview of conventional antifungal drugs, their mechanisms of action, and resistance development while also exploring newly approved treatments and those in clinical trials. Additionally, it examines innovative strategies being researched to combat fungal resistance, including multi-drug resistance. Here the following major and minor comments below:
Title: The title appears exaggerated and could benefit from a more scientifically objective tone. The overuse of buzzwords such as "Revolutionizing," "Cutting-Edge," and "Game-Changing" reduces its clarity and impact. Consider rewording it for precision and conciseness.
Response: We appreciate the reviewer’s valuable feedback and agree that a more scientifically precise title would enhance the manuscript’s clarity and impact. We have carefully revised the title to ensure it remains professional, objective, and reflective of the manuscript’s scope. The new title, "[Beyond Conventional Antifungals: Combating Resistance through Novel Therapeutic Pathways]
Abstract: The abstract is wordy, lacks clarity, and would benefit from a stronger conclusion. There is redundancy in phrasing, and a more structured flow with precise language would enhance readability. A final sentence reinforcing the significance of novel strategies and future research directions in antifungal treatment is recommended for better impact.
Response: We appreciate the reviewer’s insightful comments and have restructured the abstract to improve clarity, conciseness, and readability.
Graphical Abstract: The quality of the graphical abstract is poor, with low pixel resolution making it difficult to interpret. A clearer, high-resolution version should be provided to enhance readability and comprehension.
Response: We appreciate the reviewer’s feedback regarding the graphical abstract. To address this concern, we have replaced the existing graphical abstract with a high-resolution version that enhances clarity and readability.
Section 1: Introduction
The introduction should first provide an overview of fungal biology, followed by their advantages and disadvantages before discussing fungal infections.
Response: We appreciate the reviewer’s insightful recommendation and have restructured the Introduction to enhance logical flow and clarity.
The application of fungi should be mentioned, and references should be cited, such as:
https://doi.org/10.1093/g3journal/jkac224
https://link.springer.com/article/10.1007/s11274-023-03737-7
Response: We appreciate the reviewer's insightful suggestion. To address this, we have also incorporated the recommended references.
The transition to the research gap needs to be strengthened to highlight the importance of the study.
Response: We appreciate the reviewer’s valuable suggestion. To improve the logical flow and emphasize the significance of the study, we have strengthened the transition between the discussion on antifungal resistance and the research gaps section.
Section 2:
Some content is redundant, particularly in discussions on immunosuppression and invasive procedures. Streamlining these sections would enhance readability. Some sentences are overly complex. Simplifying them into shorter sentences would make the text easier to follow.
Response: We appreciate the reviewer’s insightful feedback. In response, we have streamlined Section 2 by eliminating redundant discussions while maintaining key points essential for understanding antifungal resistance.
Figures 1 & 2:
The graphical quality of Figures 1 and 2 is poor. Please improve the resolution and clarity.
Response: We appreciate the reviewer’s feedback. To address this concern, we have replaced the existing graphical abstract with a high-resolution version that enhances clarity and readability.
In Figure 2, the fungus is depicted as yeast budding. However, fungi have various morphological forms that infect humans. Consider including multiple fungal morphologies.
Response: We appreciate the reviewer’s observation regarding Figure 2. However, our intention was not to depict yeast budding but rather to illustrate the fungal cell division process in the context of griseofulvin's mechanism of action. The large circle represents the fungal cell, while the smaller circle signifies the daughter cell formed via mitosis.
Section 3:
A summary table comparing antifungal drugs, their mechanisms of action, and clinical applications would improve readability and comprehension.
Response: We appreciate the reviewer’s suggestion. In response, we have added a comprehensive table summarizing the major antifungal drug classes, their mechanisms of action, and clinical applications.
Some statements require more recent references, particularly those discussing resistance mechanisms and emerging antifungal treatments.
Response: We appreciate the reviewer’s suggestion and agree that incorporating recent references will strengthen the manuscript. In response, we have updated citations.
Section 4:
This section provides good coverage of resistance mechanisms; however, a brief summary at the end of each subsection would reinforce key takeaways.
Response: We appreciate the reviewer’s suggestion. In response, we have added a concise summary at the end of each subsection.
Section 5:
The manuscript lists various synthetic and natural antifungal drugs but lacks a comparative analysis. A table comparing their efficacy, side effects, and resistance potential would be beneficial.
Response: We appreciate the reviewer’s insightful suggestion. In response, we have added a comparative table in Section 5 that provides a detailed analysis of the synthetic and natural antifungal drugs discussed.
Section 6:
Some sentences are unnecessarily complex. Simplifying these sentences by breaking them into smaller parts will improve readability.
Response: We appreciate the reviewer’s valuable feedback. In response, we have simplified unnecessarily complex sentences in Section 6 to improve readability.
The manuscript inconsistently refers to fungal species, alternating between full names (e.g., Candida albicans) and abbreviations (C. albicans). Standardizing the naming format would enhance clarity.
Response: Thank you for your careful review. We have standardized the naming format for fungal species throughout the manuscript, ensuring consistency.
The discussion on antifungal lock therapy (Lines 484–505) mentions ethanol-based solutions but lacks alternative approaches. Including ongoing clinical trials or alternative solutions would strengthen this section.
Response: We appreciate the reviewer’s valuable suggestion. In response, we have expanded the discussion on antifungal lock therapy to include alternative solutions such as taurolidine and heparin-based antifungal combinations. Additionally, we have incorporated information on ongoing clinical trials evaluating novel lock therapies.
Minor grammatical errors and awkward phrasing are present. For example, "antifungal medications can be hazardous to the host because fungi are eukaryotic" (Line 415) could be reworded for better clarity.
Response: We appreciate the reviewer’s observation regarding grammatical errors and awkward phrasing. In response, we carefully reviewed and revised the manuscript to improve clarity, grammar, and sentence structure.
New Section Recommendation: Enzymes Involved in Antifungal Resistance
A new section should be added discussing enzymes that contribute to antifungal resistance.
The section should cover key enzymes, their role in resistance mechanisms, and potential therapeutic targets.
Response: We appreciate the reviewer’s suggestion to include a section on enzymes involved in antifungal resistance. We would like to clarify that these key enzymes, including ERG11, FKS1, FKS2, and Cytochrome P450, are already discussed in Section 4, 'Development of Resistance to Conventional Antifungal Drugs.
The following references should be cited:
http://doi.org/10.1155/2017/9870679
https://doi.org/10.1016/bs.mie.2023.12.016
Response: We appreciate the reviewer’s suggestion to include these references. We have reviewed the suggested papers and incorporated them appropriately into relevant sections of the manuscript to strengthen our discussion.
Section 7: Conclusion
The conclusion provides a comprehensive overview of antifungal resistance but should include a structured comparison between novel and conventional treatments in terms of efficacy and safety.
A dedicated section on future research needs and emerging therapeutic trends would significantly enhance the review.
Potential areas for further exploration, such as alternative drug delivery systems, combination therapies, and host-directed strategies, should be highlighted.
Response: We appreciate the reviewer’s suggestion to enhance the conclusion. In response, we have restructured the conclusion.
A final section titled 'Future Directions and Final Remarks' should be included to summarize key takeaways and outline promising research avenues in antifungal therapy.
Response: We appreciate the reviewer’s suggestion. In response, we have added a new section titled 'Future Directions and Final Remarks,'
Comments on the Quality of English Language
The section's English could be clearer and more concise to better convey the research. Sentences are often long and complex, making readability difficult. Improved structure, reduced redundancy, and stronger transitions would enhance clarity.
Response: We appreciate the reviewer’s recommendation to improve the manuscript's clarity and conciseness. In response, we have simplified complex sentences, enhanced transitions, and reduced redundancy throughout the text.

Reviewer 3 Report
Comments and Suggestions for Authors
This review provides a general snapshot of the conventional antifungal drugs and their mechanisms of action and development of fungal resistance and highlights the newly approved drugs and those that are still in clinical trials. Furthermore, it discusses in depth the up-to-date studies that employ novel approaches to combat fungal resistance and even multi-drug resistance. Overall, its good effort by the authors but I have few suggestions for the improvement of this review.
Abstract: Abstract is fine enough and concise
Introduction & Literature Review:
1. Introduction can be improved. The specific objective of the review must be added in the introduction.
2. Literature review should be added with more recent works. Too much older reference should be replaced by newer one.
3. Please check the fungal names, it must be italic throughout the manuscript.
4. The discussion is thorough but could be more focused. The authors should address the mechanistic insights and future research directions.
5. The limitations of the study are briefly mentioned but could be expanded.
6. The basic information of the drugs can be summarized in the form of table.
Images and artwork:
1. All images and artwork are blur. Please provide the clear images.
2. Full form of the abbreviation must be added in Images caption.
Conclusion: It sounds incomplete. Future prospect should be added in the conclusion

English is fine
Author Response
Reviewer 4
This review provides a general snapshot of the conventional antifungal drugs and their mechanisms of action and development of fungal resistance and highlights the newly approved drugs and those that are still in clinical trials. Furthermore, it discusses in depth the up-to-date studies that employ novel approaches to combat fungal resistance and even multi-drug resistance. Overall, its good effort by the authors but I have few suggestions for the improvement of this review.
Abstract: Abstract is fine enough and concise
Introduction & Literature Review:
- Introduction can be improved. The specific objective of the review must be added in the introduction.
Response: We appreciate the reviewer’s valuable suggestion. To address this, we have incorporated a specific objective at the end of the Introduction.
- Literature review should be added with more recent works. Too much older reference should be replaced by newer one.
Response: We appreciate the reviewer’s suggestion. In response, we have updated the literature review by incorporating more recent studies (2020–2024).
- Please check the fungal names, it must be italic throughout the manuscript.
Response: We appreciate the reviewer’s observation. We have carefully reviewed the manuscript and ensured that all fungal species names are consistently italicized throughout the text, figures, tables, and references where applicable.
- The discussion is thorough but could be more focused. The authors should address the mechanistic insights and future research directions.
Response: Thank you for your thoughtful feedback. To improve focus, we have refined the discussion section by emphasizing key mechanistic insights related to antifungal resistance.
- The limitations of the study are briefly mentioned but could be expanded.
Response: Thank you for your suggestion. I have expanded the discussion on the limitations of the study to provide a more comprehensive assessment.
- The basic information of the drugs can be summarized in the form of table.
Response: Thank you for your helpful suggestion. I have now summarized the basic information on the drugs in a table to enhance clarity and readability.
Images and artwork:
- All images and artwork are blur. Please provide the clear images.
- Full form of the abbreviation must be added in Images caption.
Response: We appreciate the reviewer’s feedback regarding the figures. All revised figures have been updated in the manuscript with high resolution.
Conclusion: It sounds incomplete. Future prospect should be added in the conclusion
Response: We appreciate the reviewer’s valuable suggestion. To address this, we have incorporated a future prospect at the end of the conclusion.

Reviewer 4 Report
Comments and Suggestions for Authors
This review covers an interesting topic and explores the growing threat of antifungal resistance as well as emerging strategies such as nanotechnology, immunomodulation, and drug repurposing to combat multidrug-resistant fungal infections. A few issues need to be addressed before acceptance can be considered.
- All the figures are not quite clear. Please revise them in a clearer version.
- A comprehensive review was focused on antifungal resistance but new insights compared to previous reviews should be highlighted.
- The review mentions nanotechnology and drug repurposing but does not critically compare these strategies with other emerging therapies.
- A schematic diagram summarizing key antifungal resistance pathways is suggested.
- A clear outline of key research gaps and propose specific areas for future investigation are missing.
Author Response
Reviewer 5
This review covers an interesting topic and explores the growing threat of antifungal resistance as well as emerging strategies such as nanotechnology, immunomodulation, and drug repurposing to combat multidrug-resistant fungal infections. A few issues need to be addressed before acceptance can be considered.
All the figures are not quite clear. Please revise them in a clearer version.
Response: We appreciate the reviewer’s feedback regarding the figures. All revised figures have been updated with a high resolution one. We hope these modifications meet the reviewer’s expectations.
A comprehensive review was focused on antifungal resistance but new insights compared to previous reviews should be highlighted.
Response: We sincerely appreciate the reviewer’s suggestion. To clearly differentiate our review from prior works, we have explicitly highlighted the novel insights throughout the manuscript.
The review mentions nanotechnology and drug repurposing but does not critically compare these strategies with other emerging therapies.
Response: We appreciate the reviewer’s insightful suggestion. To strengthen the discussion, we have added a dedicated comparative table under "New Approaches and Strategies to Combat Fungal Resistance," critically evaluating nanotechnology and drug repurposing in terms of mechanism, efficacy, limitations, and clinical applicability.
A schematic diagram summarizing key antifungal resistance pathways is suggested.
Response: Thank you for your valuable suggestion regarding the inclusion of a schematic diagram summarizing key antifungal resistance pathways. We would like to highlight that Figure 3, presented in the section "Development of Resistance to Conventional Antifungal Drugs," already serves this purpose. This figure comprehensively illustrates the major mechanisms of antifungal resistance, including azole resistance, echinocandin resistance, amphotericin B resistance, and biofilm-associated resistance, by depicting key molecular changes such as ERG11 and FKS mutations, efflux pump overexpression, and alterations in drug binding sites.
However, to further enhance its clarity, we have refined the figure by improving text readability and contrast, ensuring that all elements are clearly visible.
A clear outline of key research gaps and propose specific areas for future investigation are missing.
Response: We appreciate the reviewer’s valuable suggestion. To address this, we have incorporated a discussion at the end of the Introduction and Conclusion that highlights key research gaps and the need for more applicable models to facilitate the translation of novel anti-fungal strategies into clinical use.

Round 2
Reviewer 1 Report
Comments and Suggestions for Authors
Dear author, thank you for your excellent work and prepared review manuscript.
The manuscript provides a broad review of research, focusing on novel therapeutic strategies and potential targets for new development. Emerging fungal pathogens and their resistance patterns that have not been extensively described in previous literature are discussed. As with all large-scale work, mistakes are bound to happen.
A list of references is provided without following the principles of uniform citation.
DOI references must be provided and included in the same form in all sources.
Several proofreading errors were found, so the manuscript should be corrected.
Author Response
Dear author, thank you for your excellent work and prepared review manuscript.
We sincerely appreciate the reviewer’s words and positive feedback on our work. It is highly encouraging to know that our manuscript was well received.
The manuscript provides a broad review of research, focusing on novel therapeutic strategies and potential targets for new development. Emerging fungal pathogens and their resistance patterns that have not been extensively described in previous literature are discussed. As with all large-scale work, mistakes are bound to happen.
A list of references is provided without following the principles of uniform citation.
Response: We appreciate the reviewer’s feedback regarding inconsistencies in reference formatting. We have carefully revised the reference section to ensure uniformity in citation style, following the journal's guidelines.
DOI references must be provided and included in the same form in all sources.
Response: We appreciate the reviewer’s observation regarding DOI consistency. We have also verified that each DOI is correctly formatted and consistently applied across all references according to the journal's citation style.
Several proofreading errors were found, so the manuscript should be corrected.
Response: We appreciate the reviewer’s careful reading of our manuscript and for pointing out the proofreading errors. We have thoroughly revised the manuscript to correct these issues and ensure clarity, consistency, and grammatical accuracy.
Reviewer 2 Report
Comments and Suggestions for Authors
All the comments and suggestions are well addressed.
Thank you authors.
Comments on the Quality of English LanguageImproved.
Author Response
We sincerely appreciate the reviewer’s words and positive feedback on our work. It is highly encouraging to know that our manuscript was well received.
Reviewer 4 Report
Comments and Suggestions for Authors
No further question from my side
Comments on the Quality of English LanguageProper English
Author Response

(The authors gave the same response as above.)
